# Problems and Prospects of Improving Abiotic Stress Tolerance and Pathogen Resistance of Oil Palm

**DOI:** 10.3390/plants10122622

**Published:** 2021-11-29

**Authors:** Lu Wei, Jerome Jeyakumar John Martin, Haiqing Zhang, Ruining Zhang, Hongxing Cao

**Affiliations:** 1Coconut Research Institute, Chinese Academy of Tropical Agricultural Sciences, Wenchang 571339, China; wl79912021@163.com (L.W.); jeromejeyakumarj@gmail.com (J.J.J.M.); zhqhzau68@163.com (H.Z.); zrnacademic9910@163.com (R.Z.); 2Hainan Key Laboratory of Tropical Oil Crops Biology, Wenchang 571339, China

**Keywords:** cold resistance, disease resistance, drought resistance, *Elaeis guineensis*, stress tolerance

## Abstract

Oil palm crops are the most important determinant of the agricultural economy within the segment of oilseed crops. Oil palm growing in their natural habitats are often challenged simultaneously by multiple stress factors, both abiotic and biotic that limit crop productivity and are major constraints to meeting global food demands. The stress-tolerant oil palm crops that mitigate the effects of abiotic stresses on crop productivity are crucially needed to sustain agricultural production. Basal stem rot threatens the development of the industry, and the key to solving the problem is to breed new oil palm varieties resistant to adversity. This has created a need for genetic improvement which involves evaluation of germplasm, pest and disease resistance, earliness and shattering resistance, quality of oil, varieties for different climatic conditions, etc. In recent years, insights into physiology, molecular biology, and genetics have significantly enhanced our understanding of oil palm response towards such stimuli as well as the reason for varietal diversity in tolerance. In this review, we explore the research progress, existing problems, and prospects of oil palm stress resistance-based physiological mechanisms of stress tolerance as well as the genes and metabolic pathways that regulate stress response.

## 1. Introduction

Oil palm (*Elaeis guineensis*), a perennial tropical oil crop, is widely distributed in tropical areas between 10° S and 15° N [1] and is mainly cultivated in Southeast Asia, West, and Central Africa, and North and Central South America [2]. In recent years, precipitation in West and Central Africa has decreased annually, resulting in a shortage of water [3,4] and affecting the cultivation of oil palm. The northern part of South America has a tropical arid climate, with perennial high temperatures and little rain, which is not conducive to the growth of oil palm [5]. In Southeastern Asia, there is a definite dry and rainy season, and the temperature drops during the dry season [6], inhibiting the growth of oil palm young fruits [7]. As a result, the low temperature and arid environment are unfavorable for oil palm growth and development, directly impacting oil palm yield [8]. Stress tolerance enables oil palm to trigger a number of protective strategies for enhancing survival and rapid recovery with full metabolic activities upon the return of favorable conditions; however, tolerance strategy is limited to oil palm species. The research on stress-resistant breeding of oil palm can improve the yield, enhance resistance, and promote the production and development of the oil palm industry, to further satisfy the demand for palm oil. Abiotic stress generates a wide range of morphological, physiological, biochemical, and molecular changes in plants, that have a negative impact on development and productivity [9]. Drought, temperatures, and oxidative stress are consistently related and may induce cellular damage [10]. For example, droughts are predominantly expressed as osmotic stress, resulting in a disturbance of homeostasis and ion distribution in the cell [11]. Oxidative stress is commonly caused by high temperatures, or drought stress, which can lead to the denaturation of functional and structural proteins [12]. Meanwhile, the disease of oil palm is another important factor that affects the normal growth and development, such as stem rot, which causes the decay of the root, base, stem, and so on. Therefore, low temperatures, drought, and disease hinder the global planting and development of oil palm, reducing production and the planting area, affecting the global demand for palm oil. Throughout the development history of oil palm, stress-resistant cultivation has always been a concern to relevant stakeholders. Traditional and modern breeding, as well as biotechnology are widely used to enhance oil palms with traits that confer stress resistance. Various environmental stresses have a significant impact on global oil palm productivity and crop quality. Functional genomics practices have contributed to our understanding of stress signal perception and transduction, as well as the underlying molecular regulatory network [13]. Several stress-inducible genes and transcription factors that regulate stress-inducible expression were revealed by using transgenic techniques. A thorough understanding of the action of oil palm resistance will lay the foundation for elucidating the physiological, metabolic mechanisms, gene expressions, and developmental activities in all aspects of oil palm life processes for improving a multitude of oil palm traits.

## 2. Morphological Changes of Oil Palm under Abiotic Stress

Severe moisture conditions (drought and floods), extreme temperatures (hot and cold), and radiation (UV, ionizing radiation) [14] all contribute to the emergence of abiotic stressors (Figure 1). Stresses such as high temperatures and precipitation are more common and are causing a significant decline in agricultural productivity [15]. During the maturation of a tree, the plants are exposed to a wide range of environmental parameters [16]. To assess the influence of abiotic stress on plants, morphological descriptors such as root diameter, root length, and root weight, leaf area, ovule, pollen variability, and anther development are typically measured [17,18,19]. Several researchers identified a substantial correlation between oil palm vegetative attributes and yield. Subronto et al. [20] showed that leaf area may be used as a selection criterion in 9-month-old oil palm seedlings in a study on seedlings of oil palm because it was significantly correlated with yield. Leaf number, seedling height, and girth, as well as seedling dry weight, were found to be strongly and positively correlated [21]. Balakrishna et al. [22] discovered a positive correlation between leaf area and bunch yield in oil palm. Marjuni et al. [23] discovered a correlation between palm height (HT) and the quantity of fresh fruit bunches, as well as the average bunch weight. Low temperatures have a severe impact on oil palm growth and yield, causing an increase in abortion, delayed vegetative growth, and bunch ripening, as well as restricting regional distribution. Corley and Tinker [24] stated that the tamatave in Madagascar has a notably low minimum temperature of approximately 18 °C for four months, resulting in a significantly seasonal yields pattern and nearly 90% of the crop collected between June and December due to abortion and a reduced sex ratio. Thirasak et al. [25] evaluate the effect of different concentrations of NaCl concentrations on oil palm growth. Oil palm cell suspension growth rate was reduced as NaCl concentrations and exposure durations were increased. The concentrations which inhibited growth rate at 50% (IC50) were clearly seen to require prolonged exposure time whereas high concentrations required short exposure time. Oil palm cell culture can increase electrolyte leakage and proline accumulation when the concentrations of NaCl increases. Increased proline content at 2 folds higher than that of control, especially the concentration over 200 mM NaCl the oil palm cells cannot survive.

## 3. Anatomical Structure of Oil Palm under Abiotic Stress

According to Luis et al. [26], the adaptability and anatomical plasticity of oil palm seedlings and pre-germinated seedlings were substantially consistent. The epidermal cells of the test-tube seedlings were thin and the midrib of the leaves consisted of 3 to 4 vascular bundles, whereas the seedlings grown in the greenhouse had thicker leaves. Low-temperature damage to leaves was reported to be reflected in a reduction in the thickness of total leaves, spongy tissue, and palisade tissue. Changes in overall leaf thickness and tissue porosity were more visible under different low-temperature treatments, which may be utilized as structural markers for identifying oil palm cold resistance. Zeng et al. [27] reported that the thickness of the adaxial cuticle of the leaves, as well as the thickness of spongy tissue, had the maximum positive and negative direct impacts on the LT50 at low temperatures in oil palm populations. As a result, the thickness of the adaxial cuticle and the thickness of the leaf sponge tissue may be used as important structural genetic variants for identifying oil palm populations and their cold tolerance. In another study, Sari et al. [28] evaluated the changes in leaf anatomical characteristics of oil palm seedlings were examined under moderate and severe drought stress. These included the length of upper and lower epidermis cells, the width of upper dermis cells, and the thickness of mesophyll. The structure of part of the leaf was altered to adapt to the arid environment.

## 4. Role of Antioxidant Defense of Oil Palm under Abiotic Stress

Proline is also one of the stress indicators for drought as it is a critical cell solute for antioxidant systems, enzyme stabilization, signal transduction, and ROS detoxification [29]. The variation of proline levels in drought stress on oil palm has also been studied, with proline levels increasing when water was lowered and decreasing when water was re-watered [30]. Foliar application of proline was beneficial under natural environmental circumstances, generating a substantial rise in carotenoids, polyphenols, chlorophyll, and proline. It has the potential to improve stress tolerance and thereby mitigate the negative consequences of drought stress [31]. Under saline conditions, the foliar shower of proline adjusts osmotic potential and has a crucial role in plant development in arable crops such as rice [32]), maize [33], wheat [34], and cotton [35]. According to Abdul Jaleel et al. [36], increased proline production improves plants in maintaining low water potential and obtaining water from the environment under stressful conditions. If the rate of antioxidant activity is lower than the rate of ROS generation, lipid peroxidation in the cell membrane exists. The peroxidation of unsaturated fatty acids in phospholipid membranes produces Malondialdehyde (MDA) as a residue [10,11]. The amount of lipid peroxidation due to free radicals can indicate oxidative damage at the cellular level [37]. Changes in membrane structure, liquidity, ion transport, enzyme activity, and protein interactions each have the potential to induce membrane damage, leading to membrane leakage [10,11]. MDA levels were considerably larger in severe stress than in control, but there was no difference in moderate stress. The MDA content was higher under moderate stress, but not statistically different from the control. The content of MDA, soluble sugar, and the relative conductivity initially increased and subsequently decreased in oil palm leaves. This result showed that plants respond to cold stress by increasing MDA, which increased substantially under severe stress. In plants, free radicals are generally detoxified by enzymatic or nonenzymatic mechanisms. Non-enzymatic antioxidants known as carotenoids seek to preserve chlorophyll from oxidative degradation. Turhadi et al. [38] used Polyethylene Glycol (PEG) as a plant drought tolerance response to trigger an increase in carotenoids in transgenic plants caused by drought. Carotenoids and the gene expression of P5CS showed a positive correlation with chlorophyll content which was similar to the result obtained in barley under drought stress [39]. Whereas another study has shown a significant negative correlation between leaf drought and chlorophyll content. As the drought level increased, the chlorophyll content of oil palm leaf tissues dropped. Meanwhile, SOD is an enzyme that participates in the enzymatic conversion of highly reactive superoxide radicals to less reactive H_2_O_2_ and O_2_ [40]. As a result, enhanced SOD activity as a free radical scavenging agent is frequently used to measure plant tolerance to abiotic stress. The content of proline, soluble protein, and POD activity all exhibited a decreasing trend at first, then an increasing trend. Low-temperature stress affects the enzyme activity of oil palm. Antioxidant content can be used as an indicator to determine the level of plant resistance in response to abiotic stress. Cao et al. [41] found that when temperature decreased, peroxidase and catalase activities were increased. The activities of SOD and POD were gradually increased with the extended treatment time.

## 5. Role of Photosynthesis Change in Oil Palm under Abiotic Stress

Droughts will become more common in the future as a result of climate change, such as increased temperatures, more variability in precipitation, and changing climatic patterns. Oil palm does typically shed leaves, but drought also weakens trees and makes them more susceptible to insects, pests, and pathogenic infestation [42]. Indeed, a variety of physiological processes respond at different plant water potential thresholds [43], hence the severity of the drought season will influence the physiological response. Gong et al. [44] observed that the chlorophyll content, variable fluorescence (Fv), maximum fluorescence (Fm), variable fluorescence, and maximum fluorescence ratio (Fv/Fm) of oil palm leaves decreased, whereas initial fluorescence (F0) and non-photochemical quenching coefficient (NPQ) increased, indicating that photosystem II (PSII) was affected by drought stress, resulting in a decrease in original light energy conversion efficiency (Fv/Fm) and photosynthesis efficiency in PSII. Furthermore, depending on the osmotic potential of the culture medium, photosynthetic pigments in oil palm plants subjected to PEG-induced severe drought stress were significantly decreased [45]. In palm plantlets cultivated under acute water deficit circumstances, the photosynthetic capacities Fv/Fm, PSII, and Pn were decreased [46]. Silva et al. [47] found that water potential, CO_2_ net assimilation rate, chlorophyll concentration, ribulose-1,5-bisphosphate carboxylase, net photosynthetic rate, and stomatal conductance of two oil palm hybrids (BRS Manicoré and BRS C 2501) showed a downward trend. The total carotenoid, ascorbic acid, and glutathione concentrations of BRS C2501 were lower than those of BRS Manicoré. Based on these, chlorophyll content and chlorophyll fluorescence parameters could be used as indicators to evaluate the drought resistance of oil palm varieties. In fact, the growth characters of oil palm seedlings have been reported as sensitive parameters. In a previous study, the leaf area was significantly reduced in plants subjected to drought stress whereas plant height did not differ significantly in drought conditions after 7, 14, and 21 days. The typical regions where oil palm is grown have a distinct wet and dry season, and it would be predicted that the plants respond differently to water stress.

## 6. Molecular Biology of Oil Palm with Cold Resistance

With the advancement of molecular biology technology, distinct oil palm varieties are being explored from the molecular perspective to identify the genes related to the cold resistance of oil palm, providing a theoretical foundation for the breeding and selection of new varieties of oil palm. For instance, EgRBP42, which was a heterogeneous ribonucleoprotein RNA binding protein derived from oil palm, might regulate post-transcriptional processes of gene expression under abiotic stress conditions to enhance oil palm tolerance to cold [48]. The DREB subfamilies contain a single DNA-binding domain that acts as regulators of abiotic stress responses. The DREB proteins interact with the DRE/CRT cis-element which is present in the promoter of genes and are involved in abiotic stresses such as cold, drought, and high salinity. EgDREB1, an intron-free nuclear localization protein in oil palm, has the highest expression under low-temperature stress [49], and functional characterization was conducted in oil palm seedlings subjected to different degrees of cold severity. Chen et al. [50] identified FAD8 genes in oil palm and showed that the regulation of plastidial -3 fatty acid desaturases impacts plant stress response: these include the up-regulation of a FAD gene in Arabidopsis thaliana by low temperature. Zhang et al. [51] identified the EgWRI promoter from oil palm and observed that when treated at a low temperature, the activity of the EgWRI1 promoter began to decline after 24 h and recovered after 48 h. In this report, the GUS activity of transgenic Arabidopsis leaves was determined under low-temperature circumstances. This revealed that the oil palm EgFAD8 and EgWRI1 promoters were sensitive to low temperatures, and could be candidate promoters for further research into oil palm cold resistance. The EgMYB gene could be used as an important candidate gene to study oil palm cold tolerance (Table 1). Under low-temperature stress, 20 EgMYB genes were up-regulated [52]. Zhou et al. [53] used XJS30 and SJ64 as materials for low-temperature acclimatization and cold treatment. The results indicated that WRKY1 and WRKY7 played a role in regulating the cold resistance of XJS30 during low-temperature acclimation. WRKY1 and WRKY7 regulated the cold hardness of SJ64, genes like WRKY22, WRKY40 and WRKY55 could regulate the cold hardness of XJS30. After extensive screening of these multiple transcriptomes, studies showed that the actin gene, the adenine phosphoribosyltransferase gene, and the eukaryotic 4A initiation promoter gene were the most stable genes at low temperatures, providing basic information about the molecular biology of cold-resistant oil palm [54]. Li et al. [55] reported that the CBF gene of oil palm was significantly correlated with other genes (COR410, COR413, ICE1-1, ICE1-2, ICE1-4, SIZ1-1, SIZ1-2, ZAT10). At 12 °C, CBF1, CBF2, and CBF3 were highly correlated with all other genes. The expression of CBF1 and CBF2 genes was also positively correlated with the content of soluble sugar and proline. Ninety-five genes belonging to the WRKY family have been identified in the oil palm genome. Most of the EgWRKY genes exhibited tissue-specific expression patterns that could be induced to express under low-temperature stress. For example, Xiao et al. [56] found the expression level of 17 EgWRKY genes (EgWRKY03, 06, 07, 11, 16, 25, 26, 28, 29, 35, 52, 59, 61, 72, 76, 80, and 88) was increased by at least 2-fold, and 6 genes (EgWRKY06, 11, 25, 61, 72, and 88) were verified by RT-qPCR. Azzeme et al. [49] reported that the DREB1 gene in oil palm is involved in drought and cold signaling pathways. The DREB1 gene from oil palm increased the expression of DRE/CRT and non-DRE/CRT genes in transgenic plain tomato under low temperature and PEG treatment.

## 7. Molecular Biology of Oil Palm with Drought Resistance

The expression of EgDREB1 in oil palm seedlings and leaves increased with the increase in drought stress, and the expression of EgDREB1 in roots only increased under mild drought stress. EGRBP42 was a heterogeneous nuclear ribonucleoprotein-like RNA-binding protein from oil palm, which might enhance the resistance of plants to abiotic stresses by accelerating protein translation. The overexpression of EgRBP42 enhanced the resistance of transgenic Arabidopsis to abiotic stresses such as drought, cold, salt and flood, and enhanced the resilience after stress [48]. Based on these, EgDREB1 and EgRBP42 could be used as important candidate proteins for drought tolerance studies of oil palm. Wang et al. [61] observed that under drought stress for 14 days, the edges and tips of leaves of oil palm seedlings were necrotic, then withered and yellowed; the number, volume, and total biomass of roots decreased. The results showed that 1293 differentially expressed genes were expressed in the root after 14 days of drought stress, and the differentially expressed genes were divided into three types: hormone regulation, metabolism, and ATP-binding cassette (ABC) transporters. Three protein networks including ion transport, active nitrogen metabolism, and nitrate assimilation, were involved in the response to drought stress. These results provided the basic information on molecular biology for oil palm drought resistance.

## 8. Cultivation Techniques of Oil Palm with Drought Resistance

Brum et al. [62] monitored the transpiration (T) of a 5-year-old oil palm plantation during the dry season (DS) and rainy season (RS) using two irrigation methods, sprinkler irrigation, and drip irrigation, with no irrigation as the control (WS). The results showed that the T of DS was higher than that of WS, which had nothing to do with the irrigation system; the transpiration rate of sprinkler irrigation and drip irrigation increased with the increase in water vapor pressure deficit (VPD), and the average soil moisture of the treatment was 23% higher than that of the control. These results proved that irrigation alleviated the water pressure of oil palm due to drought stress, increased the fruit yield of oil palm, and emphasized the potential feedback of increased transpiration on water balance and hydrological regulation. Safitri et al. [63] used CROPWAT8.0 software to analyze the climatic conditions and soil properties of oil palm nurseries, simulate crop water demand, actual water consumption, and calculate the water demand of oil palm nurseries. It was shown that the average water requirement of oil palm nurseries was 3.4 mm/d. According to the daily soil water availability, plant root water retention rate, rainfall, and oil palm planting years, an average of 2.2 mm/d of net irrigation schedule and 6 mm/d of the total irrigation schedule provided basic irrigation data for the drought-resistant cultivation of oil palm. Nurwahyuni et al. [64] exposed that applying calcium to oil palm seedlings under drought stress would reduce Abscisic acid (ABA) content while increasing proline content, nitrate reductase activity, and photosynthetic rate, thereby preventing the complete closure of stomata, maintaining CO_2_ diffusion, inhibiting the accumulation of free radicals, and reducing drought damage. When the calcium application was 0.04 g/plant, the sponge cell length of oil palm seedling leaves increased, the width of the hypodermis and the diameter of the duct phloem of oil palm seedling leaves increased by 0.08g/plant, and the diameter of the xylem of oil palm seedling leaves increased by 0.12 g/plant [65], which provided basic information on fertilization for the drought-resistant cultivation of oil palm.

## 9. Research on Disease Resistance of Oil Palm

The disease usually causes about 10% to 30% yield loss in the crop production process, and it might lead to no harvest when severe [66]. The devastating diseases of oil palm, including basal stem rot, fatal yellowing, blast disease, and Fusarium wilt affect the normal growth and development of oil palm. The research on oil palm disease resistance mainly focused on pathogens, detection techniques, biocontrol bacteria, medicaments, cultivation techniques, and molecular biology.

### 9.1. Pathogens of Oil Palm Disease

*Ganoderma boninense* is a white-rot fungus that degrades lignin. The main symptoms are leaf chlorosis, plant lodging, holes in the basal stem, the rot of the root, base, and upper stem, commonly known as oil palm basal stem rot (BSR) [67]. The disease severity index (DSI) of oil palm seedlings infected with *Ganoderma boninense* was linearly negatively correlated with the chlorophyll content. In this case, the use of sensors linked to photosynthesis/chlorophyll was fluorescence, thermal or spectral sensors (multi or hyper) could be a selection tool. For instance, the spectral-based classification approach has important implications for disease management of oil palms, particularly the *Ganoderma* infection. Shafri and Anuar [68] identified sensitive wavelengths to different *Ganoderma* damage types with derivative spectra. Recently, artificial neural networks (ANN) were utilized to detect different *G. boninense* damage levels in oil palm [69]. In the early stage of the disease, the chlorophyll value showed a downward trend, but the DSI value showed no significance. Therefore, estimation of the chlorophyll content in the leaves were measured using a SPAD chlorophyll meter and could be used as a physiological index to evaluate the occurrence of diseases in oil palm seedlings [70]. Bud or spear rot (BR) disease is the most destructive disease of oil palm in South America comprising Ecuadorian Amazon, Brazil, southwest Colombia, and Suriname. The disease also has been reported in India and parts of Africa [71]. Bud rot in oil palms can manifest differently depending on the prevailing environmental conditions. In the Ecuadorian Amazon, Brazil, southwest Colombia, and Suriname, BR is mostly lethal to oil palms, while in the drier eastern plains of Colombia, it is often non-lethal, and a high rate of recovery is observed [72]. Oil palm bud rot (BR) is caused by *Phytophthora palmivora* [73]. The rot spreads from the oil palm’s meristem to the undifferentiated leaves, impeding the formation of new leaves and inhibiting the development of typical healthy leaves [74]. The quantum efficiency (ΦPSII) and maximum fluorescence intensity (Fm) of PSII, variable fluorescence yield (Fv), Fv/Fm, and Fv/F0 dropped as the severity of BR increased, whereas F0 and leaf temperature increased. ΦPSII, Fm, F0, and leaf temperature are possible early indicators of BR [73]. Fusarium wilt is caused by *Fusarium oxysporum elaeidis* (FOE). The typical internal symptom is the browning of vascular bundles, which are divided into acute Fusarium wilt and chronic Fusarium wilt. The symptom of acute Fusarium wilt is that the leaves quickly dry and die, and the trunk remains upright for 2 to 3 months before being blown off by the wind; the symptom of chronic Fusarium wilt is that the plant is short and grows slowly [75]. The sensitivity of oil palm to Fusarium wilt is polymorphic, and there are significant differences in the incidence of Fusarium wilt in different agro-ecological areas [76]. Lethal wilt (LW) is caused by Phytophthora palmivora [73]. The symptoms are dry or wet rot on arrow leaves and the yellowing symptoms of young leaves appear in the rainy season and disappear in the dry season.

Oil palm seedlings infected by *Pythiurn splenderns* have gray or brown-black spots on the stems, and the whole plant is dehydrated and wilted and eventually dies. Chen et al. [77] isolated and identified a Pythium splendens Braun strain from soil samples collected from Bawangling Nature Reserve, Hainan. It has the potential to induce typical blast disease symptoms. The African oil palm ringspot virus (AOPRV) causes leaf mottle disease. Yellow spots occur initially on the leaves, then the leaves turn dry and the plant eventually dies, and ultimately, purple circular and necrotic patches emerge on the stems [78]. Zheng et al. [78] found that the most common disease was leaf mottle, and there were six pathogens, including *Phyllosticta*, *Colletotrichum*, *Alternaria*, *Pseudomonas, Phomopsis*, *Phoma*, and *Pestalotiopsis*. The field pathogenicity of the six pathogens was determined, and it was found that the lesions caused by Colletotrichum and Alternaria were smaller and the pathogenicity was weak, which was speculated to be saprophytes. *Phyllanthus*, *Pseudostem, Phoma*, and *Polychaete* could infect oil palm varieties RYL31 simultaneously, and *Phyllosporium* could also infect oil palm varieties RYL14 and RYL33, and it is speculated that this pathogen was more parasitic. The optimum growth temperature and pH of *Phoma herbarum* and *Phomopsis* sp. in Deuteromycotina were 20~25 °C and 25 °C as well as 6.0~8.0 and 7.0, respectively. PDA medium and PSA medium were suitable for the growth of *Phoma herbarum*, and PDA was the most suitable for the growth of Phomopsis sp. in *Deuteromycotina*. The lethal temperature of *Phomopsis* sp. in *Deuteromycotina* was 50 °C [78]. Three fungal pathogens were respectively inoculated on 24 different healthy oil palm leaves by the method of needle inoculation. According to the size of the diseased spots on the leaves, the resistance of 24 oil palm varieties to pathogens was determined. It was found that the oil palm varieties with strong resistance to YZ-4 were RY-2 and RY-6. The oil palm varieties with strong resistance to YZ-6 were RY-9, RY-14, RY-15, RY-31, RY-37, and RY-38. The oil palm varieties with stronger disease resistance to YZ-8 are RY-38. It can be seen that there are differences in the resistance of different varieties to pathogenic bacteria.

### 9.2. Detection Technology for Oil Palm Disease Resistance

The detection technology of oil palm base rot mainly includes extraction methods and model establishment, which is conducive to early detection and control of the disease and guarantees the growth and development of oil palm. The DNA extraction method combined with a disposable microfluidic device and a biosensor realizes the automated, high-throughput, label-free detection of the pathogens of oil palm base rot [79]. There is a good correlation between ergosterol and the degree of infection of oil palm. The microwave-assisted extraction (MAE) method can effectively extract ergosterol from oil palm roots of grade 2, 3, and 4 diseases. It is suitable for the detection of oil palm BSR disease in the field [80]. There are significant differences between feature selection models (SVM-FS, RF, and GA), electrical characteristics (impedance, capacitance, dielectric constant, and dissipation factor), and the degree of oil palm susceptibility (healthy, mild, moderate, severe). Among them, the SVM-FS model has the best prediction effect, with an average accuracy rate of 74.77%. The prediction results of RF and GA have high accuracy. SVM-FS, RF, and GA can be used as the most suitable feature selection models for detecting BSR disease [81]. The Two-band enhanced vegetation index 2 (EVI2) is the best spectral index for detecting orange spot diodel and is most suitable for predicting, classifying, and locating oil palm diseases caused by cotton bollworms. This model has the potential to detect BSR early and distinguish the severity of oil palm diseases [82]. BSTRACT field spectroscopy is a fast and non-destructive analytical tool for assessing plant adversity and disease. The ratio 3 is the best spectral index for early detection of BSR disease in oil palm seedlings because this index has the highest average silhouette width (ASW) value, which is highly correlated with the total leaf chlorophyll (TLC) of the healthy oil palm, and is an important indicator reflecting the vital status of oil palm [83].

By analyzing the reflectance spectra of oil palm seedlings and healthy oil palm seedlings inoculated with coconut cadang–cadang viroid (CCCVd), it was shown that the infrared inflection point (ReIP) at 700 nm could well reflect the condition of oil palm seedlings under CCCVd stress. The dual-band enhanced vegetation index (EVI2) is the best spectral index for detecting orange spot disease in mature oil palm forests [84]. Zhang et al. [85] used a loop-mediated isothermal amplification technique to develop a rapid and sensitive detection method for oil palm damping-off bacteria and designed a total of 6 primers based on 8 loci in the ITS region gene. The results showed that the primers had good specificity for the detection of oil palm damping-off bacteria; the minimum detection limit of primer sensitivity was 0.12 ng·μL^−1^ which was 100 times higher than the ordinary polymerase chain reaction (PCR); the detection results could be observed with the naked eye after testing with SYBR Green1. Nababan et al. [86] used the naive Bayes method to design an intelligent application program with input and output planning, which could diagnose the types of oil palm plant diseases.

### 9.3. Biocontrol Bacteria, Fungicides and Cultivation Techniques of Oil Palm Disease Resistance

The current control strategies for pathogen infection are focused on long-term and short-term control by keeping disease incidence below the economic threshold of infected palms. Short-term control strategies include the routine roguing of infected stands, fungicide intervention, and the use of biocontrol agents, whilst long-term strategies include breeding for resistance. Both have been hampered by a lack of understanding of disease mechanisms and host-pathogen interactions, which are associated with factors of pathogenicity, resistance, tolerance, and susceptibility. Cao et al. [87] found that the biocontrol bacteria inhibited *Ganoderma lucidum* much more effectively under potato medium (PD) liquid culture conditions than under potato agar medium (PDA) solid culture conditions, indicating that the biocontrol bacteria could fully contact and act on the pathogenic mycelium in PD liquid culture to achieve the best antagonistic effect. Oil palm seedlings treated with *Streptomyces palmitos* CMU-AB204^T^ had the highest plant vigor and inhibitory activity against *Ganoderma lucidum*. Therefore, CMU-AB204^T^ was a promising biocontrol bacteria that could be used to protect oil palm trees from BSR disease [88]. The endoparasitic *Trichoderma* strain had the best control effect on oil palm seedling leaf mottle disease because it could inhibit the growth of hyphae in vitro, reduce disease symptoms in the body and the field, and increase the activity of phenylalanine ammonialyase (PAL), peroxidase (POD) and polyphenol oxidase (PPO) in oil palm leaves [89].

Nano-carrier capsules (polymers, lipids, carbon-based materials, and metals) could be loaded with active pesticide formulations. It has become a sustainable alternative for developing a low-toxic and effective pesticide nano-drug delivery system and improving the safety of the oil palm industry [90]. Chitosan-based agricultural fungicides have been used as sustainable options for root rot management. Chitosan, in particular, is often used to improve plant defense mechanisms and natural immunity, preventing pathogens from infecting plants [91]. To detect pest and disease outbreaks as early as possible, the plantation should be monitored at least every two months. Table 2 summarizes the most prevalent oil palm pests and diseases. Insecticides carbaryl, deltamethrin, fipronil, and imidacloprid have the potential to significantly enhance the mortality of the South American palm weevil and hence control the population of oil palm trees [92]. The endophytic *Bacillus subtilis* preparation could delay the incubation period of oil palm seedlings and reduce the disease intensity, and had no significant effect on the plant height, leaf number, diameter, root volume, root-to-shoot ratio, and root dry weight of the seedlings [93].

Rebitanim et al. [94] studied a new fertilizer technology to promote the growth and development of oil palm seedlings, reduce the severity of oil palm stem basal rot, the incidence and mortality of seedlings, and reduce the disease, which proved that the nutrients in GanoCare^®^ allowed the seedlings to establish a strong defense system. Foliar spraying of mineral nutrients (calcium, copper) and salicylic acid (SA) could improve the control effect of root rot of oil palm seedlings [95]. When the soil pH was 6.0, it could reduce the infection of *Ganoderma lucidum* on oil palm seedlings [96]. The incidence of oil palm stalk rot in high-density oil palm forests was significantly higher than that of low-density oil palm forests [97]. Therefore, the planting density of oil palm forests should be planned to reduce the incidence of oil palm stalk rot.

### 9.4. Molecular Biology of Oil Palm Disease Resistance

Screening for resistant varieties for disease control are difficult with a confirmed pathogen as no standard protocols exist for the infection and limited understanding of the spread of the disease could result in false identification of possible resistant varieties [98]. Nevertheless, there are several hybrids between *Elaeis oleifera* × *E. guineensis* Jacq. have been identified as a potential for disease resistance which expresses less severe symptoms and a slower progression of the disease among infected tissue [99]]. In terms of molecular biology, research on oil palm disease resistance genes has contributed to the cultivation of high-yielding oil palm varieties. Hanin et al. [100] identified that the oil palm embryogenic callus had a construct with AGLU1 and RCH10 genes and a construct with a Basta^®^ resistance gene (bar); after selection with herbicides, polymerase chain reaction (PCR) showed that the ratio 8/25 plants were positive for the presence of three transgenes (bar, AGLU1, and RCH10 genes); quantitative reverse transcription PCR (RT-qPCR) showed that five of the eight plants expressed AGLU1 and RCH10 genes; these five strains were infected by *Ganoderma lucidum* and two strains showed resistance. Using the alfalfa glucanase (AGLU1) gene and rice chitinase (RCH10) gene to biologically mediate oil palm, it could be cultivated as an anti-fungal oil palm. The pathogenicity of *G. boninense* was related to the cell wall degrading enzymes (CWDEs) released during the saprophytic and necrotrophic stages of infection of the oil palm host. Ramzi et al. [101] successfully identified the cell wall degrading enzymes in the NJ3 gene of *Ganoderma lucidum angustifolia* by analyzing the genome, and using plant-host interaction (PHI) to identify several genes related to cell wall degrading enzymes, including polysemi lacturonidase and laccase. These enzymes degraded plant cell walls and promoted the pathogenesis of fungi. These findings provided basic knowledge about the genetic ability of fungi and the ability of cell wall degrading enzymes. Nascimento et al. [102] used a two-dimensional liquid chromatography-tandem mass spectrometry (2D-UPLCMSE) analysis method to study the changes in oil palm protein components affected by fatal yellowing. The expression levels of transketolase, isoflavone reductase, cinnamyl alcohol dehydrogenase, caffeic acid 3-O-methyltransferase, adenosylmethionine synthase, acetaldehyde dehydrogenase, and ferritin gradually increased during the development of the disease, indicating that these proteins played an important role in making oil palm resistant to fatal yellowing.

## 10. Problems and Prospects of Oil Palm Resistance

Many studies have been conducted on the cold resistance of oil palm, including physiology, anatomy, and molecular biology. Further investigation into the cold resistance mechanism and genetic process of oil palm is required. It will provide a theoretical foundation for producing novel cold-resistant cultivars by conducting a comprehensive study on the cold resistance mechanism. The regulation mechanism of plant hormones on oil palm under drought stress has not yet been investigated. The above-mentioned oil palm responses to abiotic stresses show that the response is complex and depends on the species and genotype, the type of stress, the length and severity of stress, the age and stage of oil palm development, the organ and cell type, the subcellular compartment, and the gene and its mode of action. The physiological and biochemical indicators, as well as anatomical structures closely related to drought resistance, must be deepened to become a drought-resistant oil palm germplasm resource, to provide accurate and easy-to-operate evaluation indicators during utilization, laying a theoretical foundation for cultivating drought-tolerant oil palm (Figure 2). The research on oil palm disease resistance focuses on four aspects: the analysis, detection, prevention, and control of pathogens, and oil palm disease-resistant cultivation techniques. There is relatively little research on disease resistance mechanisms. Oil palm resistance should be studied from a molecular perspective to provide a basis for cultivating new disease-resistant oil palm varieties. The research on the resistance of oil palm is still in the exploratory stage, and the theory of mechanism and physiological indicators has still not been refined. The mechanism of resistance to the cultivation of oil palm should be explored from a broad perspective. In recent years, oil palm breeding for stress resistance has been applied to DNA molecular markers including SSR, ISSR, SRAP, and TRAP, which greatly shortens the breeding period, facilitates the development of oil palm genetic maps, and accelerates the cultivation of new varieties of oil palm. At the same time, organogenesis, somatic embryo induction, and somatic embryo regeneration in tissue culture technology can shorten the breeding period of oil palm, reduce costs, improve breeding efficiency, and expand oil palm germplasm resources.

## 11. Conclusions

Scientists should combine biotechnology with scientific research, actual production and development in accordance with the natural environment and cultivation conditions in various regions of the world, to develop diverse high-yielding and novel oil palm varieties for planting in different regions, thereby promoting the oil palm industry, production and development, and satisfying the demand for palm oil. To fulfill the task of sustainable future food demand, oil palm crops need to maintain high yields under marginal environmental conditions and unpredictable changing climates. Oil palm crops of the future likely need to be stacked with multiple desired traits for higher yields while mitigating the effects of abiotic stresses and complex pathogenesis incidences triggered by changing climates. Therefore, understanding the full picture of how oil palm responds to stress will enable the determination of key processes that contribute to crop yield under stress. This knowledge will be valuable for plant breeding towards improving crop yield stability under changing climatic conditions.

## Figures and Tables

**Figure 1 plants-10-02622-f001:**
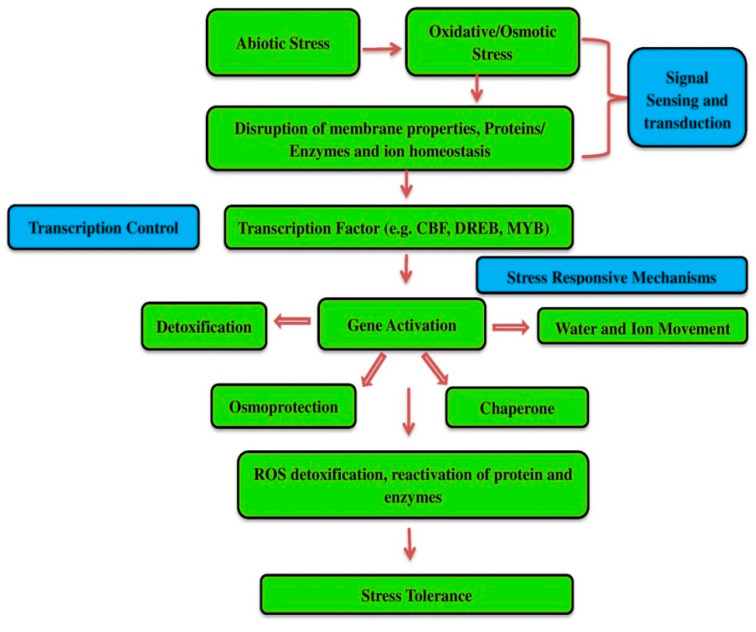
General scheme of Abiotic stress response and adaptation in plants.

**Figure 2 plants-10-02622-f002:**
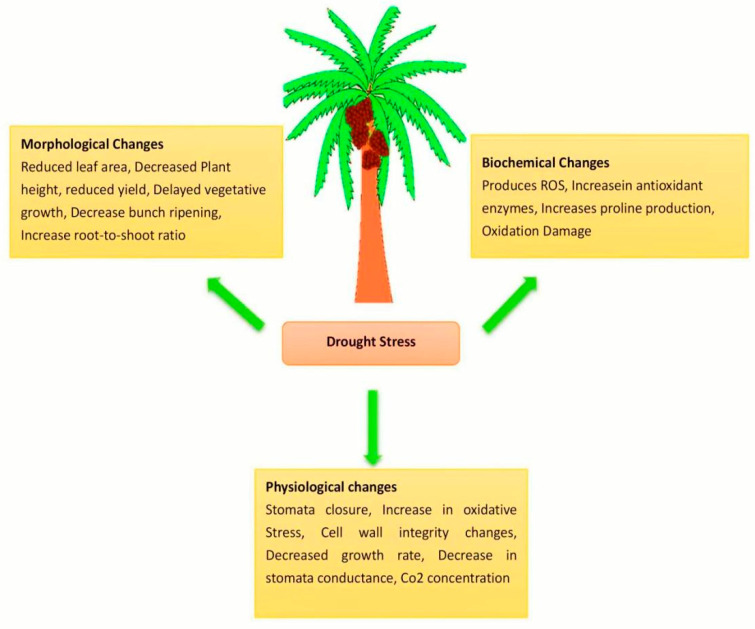
Drought tolerance in oil palm. Drought stress causes significant changes in morphological, biochemical, and physiological.

**Table 1 plants-10-02622-t001:** Summary of Abiotic stress-related genes in oil palm under different types of stress conditions.

Family	Gene Name	Types of Stress Condition	Degree/ Dose	Function	Reference
MYB	EgMYB38, EgMYB43, EgMYB57, EgMYB76, EgMYB82, EgMYB91, EgMYB104, EgMYB106, EgMYB111, EgMYB127, EgMYB133, EgMYB146, EgMYB 151, EgMYB155	Cold, Drought, and Salt	NA	Up-regulated under all abiotic stress conditions (cold, drought, and salt).	[52]
WRKY	EgWRKY18, EgWRKY64	Cold	8 °C	Involved in cold stress and negative regulator of cold response.	[56]
	EgWRKY07, EgWRKY52	Salt	400 mmol/L of NACL	WRKY gene was strongly induced and up-regulated gene in leaves after Salt stress.	[56]
AP2/ERF/RAV	EgAP2.15, EgAP2.34, EgERF23, EgERF104, EgERF130	Cold	8 °C	Increase expression of AP2/ERF genes in re-sponse to cold exposure.	[57]
	EgAP2.09, EgERF26, EgERF90, EgER104	Drought	NA	Drought stress-induced AP2 and ERF genes.	[57]
	EgERF14, EgERF73, EgRAV02	Salinity	300 mmol/L of NACL	Salt stress were induced and upregulated by ERF/RAV gene members.	[57]
bZIP	EgbZIP1, EgbZIP4, EgbZIP27, EgbZIP44, EgbZIP52, EgbZIP68, EgbZIP77, EgbZIP85, EgbZIP86, EgbZIP89, EgbZIP95	Cold, Salt, and Drought	NA	The bZIP genes were up-regulated in response to cold, salt, or drought stress, suggesting that EgbZIP plays a significant role in stress response.	[58]
ARF	EgARF4, EgARF5, EgARF6, EgARF9, EgARF10, EgARF12, EgARF13, EgARF15, EgARF21, EgARF22	Cold (Up-regulated)	8 °C	Different types of abiotic stresses can induce the expression of EgARFs (cold, drought, and salt). The ARF gene functional investigations in oil palm and serve as a genetic resource platform for oil palm abiotic stress resistance breeding.	[59]
	EgARF1, EgARF3, EgARF8, EgARF14, EgARF17, EgARF18, EgARF19, and EgARF20	Cold (Down-regulated)			
	EgARF4, EgARF6, EgARF9, EgARF10, EgARF12, EgARF13, EgARF15, EgARF16, and EgARF22	Drought (Up-regulated)	NA		
	EgARF1, EgARF14, EgARF17, EgARF18, EgARF19, EgARF20, and EgARF21	Drought (Down-regulated)			
	EgARF9, EgARF10, EgARF17, and EgARF22	Salt (Up-regulated)	300 mmol/L		
	EgARF3, EgARF4, EgARF5, EgARF8, EgARF14, EgARF15, EgARF16, EgARF18, EgARF19, EgARF20, and EgARF21	Salt (Down-regulated)			
LEA	EgLEA4	Drought	NA	Enhance drought tolerance.	[60]

**Table 2 plants-10-02622-t002:** Common pests and diseases of oil palm and Management.

Disease	Symptoms	Control
Spindle Bug	Necrotic sores and dry ground spots on leaves spindle fail to open	Keep perforated poly sachets loaded with porate (2 g) in the leaf axil.
Tussock Caterpillar	Defoliation of leaves	Damaged leaves should be cut and burned. If the infestation is severe, spray monocrotophos (0.36%) or carbaryl (0.1%).
Root Grubs	Sudden death of young plants	Fill the seedling bags with soil that is free of root grubs. While planting the sprouts, apply 50 gms of phorate per seedling.
Termites	Hindered growth of the plant	Destroy termite mounds and drench with chlropyriphos (0.5%).
Lesser Bandicoot, *Bandicota bengalensis*	Destruction of apical region	In a suitable bait station, ideally composed of earthen pots, anticoagulant baiting with bromadiolone (0.05%) may be injected.
Wild Boar	Destruction of boll region	Wild boar scaring device may be utilized.
Rhinoceros beetle (*Oryctes rhinoceros* L.)	The leaf silhouette has “V” shaped gaps.	Damaged and dead palms, as well as decaying bunches, must be removed from the orchard. To prevent egg-laying, apply “tar” to wounds and cuts on the stem section. Use logs or pheromone baits to catch the adult beetles. Maintain a clean environment in the orchard. Trunk injection of carbaryl % WP at 1% or endosulfan 35 EC at 0.1%.
Red Palm Weevil (*Rhynchophorus ferrugineus Oliver*)	Palm wilts and dries gradually. Grubs feeding inside the trunk generate a distinctive sound.	
Case Worm	Sporadic defoliation	Spray carbaryl (0.1%) on infected leaves. If the infestation is severe, root feeding with monocrotophos may be practiced.

## Data Availability

Not applicable.

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
