# Peer review of "Problems and Prospects of Improving Abiotic Stress Tolerance and Pathogen Resistance of Oil Palm"

_plants, 2021, doi:10.3390/plants10122622_

Round 1
Reviewer 1 Report
Comments: Oil palm in the environments in which it is cultivated has suffered biotic and abiotic limitations, and a review detailing the advances would be very convenient to create a criterion for solving these stresses.
The traditional improvement strategy even with genetic gain by genomics (shown in figure 1) is medium/long term; however, it is necessary to explore, in this review the possibility of using soil and plant management that minimize these stresses, sometimes instantly. We could indicate these managements as well as gypsum in the case of soils with aluminum toxicity, foliar application of proline as an osmoregulator or growth regulators that usually stimulate roots and irrigation efficiency.
In the specific case of obtaining drought and salinity tolerant genotypes, this result could make the oil palm more viable in northeastern Brazil, which is a semi-arid region with problems of water affected by salts.
In the case of biotic limitations, the biological and chemical control of diseases with low toxicity products was explored in the review.
More detais in attached file.

Author Response
Comments: Oil palm in the environments in which it is cultivated has suffered biotic and abiotic limitations, and a review detailing the advances would be very convenient to create a criterion for solving these stresses.The traditional improvement strategy even with genetic gain by genomics (shown in figure 1) is medium/long term; however, it is necessary to explore, in this review the possibility of using soil and plant management that minimize these stresses, sometimes instantly. We could indicate these managements as well as gypsum in the case of soils with aluminum toxicity, foliar application of proline as an osmoregulator or growth regulators that usually stimulate roots and irrigation efficiency.In the specific case of obtaining drought and salinity tolerant genotypes, this result could make the oil palm more viable in northeastern Brazil, which is a semi-arid region with problems of water affected by salts.In the case of biotic limitations, the biological and chemical control of diseases with low toxicity products was explored in the review.
Reply: Thank you very much for your comments. We have done the modification according to the comments. The manuscript has been reorganized to make the expression clearer. The figures have been revised and modified according to the comments.
Reviewer 2 Report
Oil palm is an important crop. Many stress researches have been conducted. The authors of this review summarized a large amount of research information. However, the English writing is very poor and not publishable. There are many problematic sentences. Here I just list a few:
Lines 11 to 12: For development of stress-tolerant crops that mitigate the effects of abiotic stresses on crop productivity is crucially needed to sustain agricultural production.
Lines 67 to 68: Stresses such as high temperatures (heat and cold) and precipitation (drought and floods) are more common and are causing significant decline in agricultural productivity.
Lines 84 to 86: Corley and Tinker [25] stated that low temperature around 18℃, which produced a strongly seasonal yield pattern, with approximately 90% of crop is harvested between June and December, owing to abortion and lower sex ratio in winter.
Lines 87to 88: Thirasak et al. [26] evaluate the effect of various NaCl concentrations on oil palm growth.
Line 89: inhibited 50%
Line 91 to 92: Oil palm cell culture can tolerate modified by electrolyte leakage and proline accumulation the increment of concentrations of NaCl the increment of concentrations of NaCl.
Lines 93 to 94: Also increased proline content at 2 folds higher than that of control, especially the concentration at 200 mM.
Lines 111 to 116
Line 147 to 148: Among GR activity reached a significant level;
Lines 154-155: Oil palm does typically shed leaves, but drought also weakens tress and makes them more the susceptible to insects attacks and the pathogens
……..
I suggest that authors ask help from commercial English editing services or from an expert with good English writing skills in this research area.
Author Response
Comment 1:- Lines 11 to 12: For development of stress-tolerant crops that mitigate the effects of abiotic stresses on crop productivity is crucially needed to sustain agricultural production.
Reply: The stress-tolerant crops that mitigate the effects of abiotic stress on crop productivity is crucially needed to sustain agricultural production.
Comment 2: - Lines 67 to 68: Stresses such as high temperatures (heat and cold) and precipitation (drought and floods) are more common and are causing significant decline in agricultural productivity.
Reply: Thanks for your suggestion. The changes were made.
Comment 3: -Lines 84 to 86: Corley and Tinker [25] stated that low temperature around 18℃, which produced a strongly seasonal yield pattern, with approximately 90% of crop is harvested between June and December, owing to abortion and lower sex ratio in winter.
Reply: Changes were made in sentence. Corley and Tinker stated that the tamatave in Madagascar has a notably low minimum temperature of approximately 18℃ for four months, resulting in a significantly seasonal yields pattern and nearly 90% of the crop collected between June and December due to abortion and a reduced sex ratio.
Comment 4: -Lines 87 to 88: Thirasak et al. [26] evaluate the effect of various NaCl concentrations on oil palm growth.
Reply: Thirasak et al. evaluate the effect of different concentration of NaCl concentrations on oil palm growth. Oil palm cell suspension growth rate was reduced as NaCl concentrations and exposure durations were increased.
Comment 5: -Line 89: inhibited 50%
Reply: The concentrations which inhibited growth rate at 50% (IC50) were clearly seen to require prolonged exposure time whereas high concentrations required short exposure time.
Comment 6: -Line 91 to 92: Oil palm cell culture can tolerate modified by electrolyte leakage and proline accumulation the increment of concentrations of NaCl the increment of concentrations of NaCl.
Reply: Thanks for your suggestion. Changes were done in the sentence “Oil palm cell culture can increase electrolyte leakage and proline accumulation when the concentrations of NaCl increases”.
Comment 7: -Lines 93 to 94: Also increased proline content at 2 folds higher than that of control, especially the concentration at 200 mM.
Reply: Also increased proline content at 2 folds higher than that of control, especially the concentration over 200 mM NaCl the oil palm cells cannot survive.
Comment 8: -Lines 111 to 116
Reply: The sentence were rewritten as “As a result, the thickness of the adaxial cuticle and the thickness of the leaf sponge tissue may be used as important structural genetic variants for identifying oil palm populations and their cold tolerance. In another study, Sari et al. evaluated the changes in leaf anatomical characteristics of oil palm seedlings were examined under moderate and severe drought stress. These included the length of upper and lower epidermis cells, the width of upper dermis cells and the thickness of mesophyll. The structure of part of the leaf was altered to adapt to the arid environment”.
Comment 9: -Lines 154 to 155 : Oil palm does typically shed leaves, but drought also weakens tress and makes them more the susceptible to insects attacks and the pathogens
Reply: Oil palm does typically shed leaves, but drought also weaken trees and makes them more the susceptible to insect, pests and pathogenic infestation.
Reviewer 3 Report
Overall oil palm review is written well; however, the manuscript structure seems bit inconsistent. Authors are encouraged to refer to other article reviews and rearranged the manuscript structure since it is bit confusing to maintain the reading flow as review idea has to be focused on the manuscript title. The idea is very important for oil palm researchers and other stakeholders but needs further work to present it in a logical order. Also, there are several issues with appropriate use of English hence authors are encouraged to have the manuscript reviewed by native English speaker. There are other comments made in the attached review report and authors are encouraged to revised the manuscript following suggested review comments.

Author Response
Comment 1: - Line 29: “on the land”, no need of this mention since this is implied automatically.
Reply: Thanks for your suggestion. The sentence has been deleted.
Comment 2: - Line 38: “However, tolerance strategy is limited to oil palm species.”, please connect this sentence with the previous sentence since it is related to it while this sentence is too short to be on its own. Authors can write it like Stress tolerance enables…favorable conditions; however, tolerance strategy is limited to oil palm species.”
Reply: Thanks for your comment. We have followed the suggestion.
Comment 3: -Lines 38-41: “research on stress-resistant 38 breeding of oil palm can improve the yield of oil palm fruit, enhance the resistance of oil 39 palm, and promote the production and development of the oil palm industry, to further 40 satisfy the demand for palm oil.”. Too redundant as use of oil palm is repetitive and I think authors can limit this repetition
Reply: Thanks for your suggestion. The redundant word has been deleted. research on stress-resistant breeding of oil palm can improve the yield, enhance resistance, and promote the production and development of oil palm industry to further satisfy demand for palm oil.
Comment 4: - Line 53: “been a topic of great concern” present it as “a concern to relevant stakeholders”
Reply: Thanks for your suggestion. The sentence has been changed.
Comment 5: - Line 54: “biotechnology,” no need of comma, please remove the same
Reply: Thanks for your suggestion. The comma has been removed.
Comment 6: - Line 55: “that confer stress resistance” this part of the sentence looks like italicized please check and correct it as needed
Reply: Correction has been made.
Comment 7: - Lines 59-60: “using techniques”, please specify what technique?
Reply: Correction has been made to transgenic techniques.
Comment 8: - Line 71: “measures” not appropriate term, please use “descriptors or traits” instead
Reply: Thanks for your suggestion. The word “measures” were changed to “descriptors”.
Comment 9: - Line 54: “re- 72 searches” looks like a typo, should be “researchers” instead
Reply: Thanks for your suggestion. Changes were done.
Comment 10: - Lines 87-94: Paragraph starting with “Thirasak et al” should be merged with previous paragraph as the last paragraph is too small.
Reply: Thanks for your suggestion. The Paragraph has been merged.
Comment 11: - Line 100: “thin,” no need of comma after this word so please correct the same
Reply: Comma were deleted.
Comment 12: -Line 118: “Proline is also one of the stress indicators for drought. Proline is a” please correct it as
Reply: Thanks for you suggestion. The correction has been made as “Proline is also one of the stress indicators for drought as it is a”
Comment 13: -Line 120: “was also studied, with” should be written as “has also been studied with”
Reply: Thanks for you suggestion. The correction has been done.
Comment 14: Line 132: “under moderate stress, but not statistically different from the control. The content of” this part of the sentence looks like italicized please check and correct it as needed
Reply: Changes were done.Thank you.
Comment 15: -Lines 145-147: “pe- 145 roxidase (POD) activity, catalase (CAT) activity, superoxide dismutase (SOD) activity and 146 glutathione reductase (GR) activity increased” please rewrite this sentence
Reply: The sentence has rewrite as “peroxidase (POD), catalase (CAT), superoxide dismutase (SOD), and glutathione reductase (GR) activities increased”. Thank you.
Comment 16: -Line 150: “in responding to” please rewrite it as “in response to”
Reply: Changes were made. Thank you.
Comment 17: - Line 155: “the susceptible to insects attacks and the pathogens” rewrite
Reply: Thanks for your suggestion. The sentence has rewritten as “susceptible to insects, pests, and pathogenic infestation”
Comment 18: -Line 176: “7, 14 and 21 days” rewrite
Reply: Rewrite as “7, 14, and 21 days”.
Comment 19: -Line 178: “to the wet” no need of the in this part of the sentence, please correct as needed
Reply: The sentence has been corrected.
Comment 20: -Lines 194-209: Please merge both paragraphs together and avoid presenting micro/mini paragraphs
Reply: Thanks for your suggestion. The paragraphs were merged.
Comment 21: -Line 285: “leaf of were” rephrase
Reply: Sentence were repharse to “leaves were”.
Comment 22: -Line 291: “BR in” do not start the sentence with abbreviation
Reply: The word BR were elaborated
Comment 23: -Lines 291-294: “In the Ecuadorian Amazon, Brazil, south west Colombia, and Suriname, BR is mostly lethal to oil palms, while in the drier eastern plains of Colombia, BR is often non-lethal, and a high rate of recovery is observed” rephrase
Reply: Thanks for your suggestion. The sentence were rephrase to “In the Ecuadorian Amazon, Brazil, south west Colombia, and Suriname, BR is mostly lethal to oil palms while in the drier eastern plains of Colombia, it is often non-lethal, and a high rate of recovery is observed”
Comment 24: - Lines 301-315: Please merge both paragraphs together and avoid presenting micro/mini paragraphs
Reply: Thanks for your suggestion. The paragraphs were merged.
Comment 25: -Line 327: “and 7.0 respectively” should be rephrased as “and 0.7, respectively
Reply: Corrections were done.
Comment 26: -Lines 421-423: “there are several hybrid material Elaeis oleifera x E. guineensis Jacq. has 421 been identified as a potential for disease resistance by expressing less severe symptoms 422 and a slower progression of the disease in infected tissue” rewrite
Reply: The sentence were written as “there are several hybrids between Elaeis oleifera x E. guineensis Jacq. have been identified as a potential sources of disease resistance which express less severe symptoms and slow progression of the disease among infected tissue”
Comment 27: -Lines 423-425: “In terms of molecular biol-423 ogy, research on oil palm disease resistance genes is conducive to the cultivation of oil 424 palm varieties with excellent traits.” highly confusing sentence please rewrite it.
Reply: The sentence were rewritten.
Comment 28: -Line 430: “AGLU1 and RCH10 gene” should be genes, please correct the same
Reply: Changes were made.
Comment 29: -Line 446: “of the disease, indicating” no need of comma in this part of the sentence, please correct the same
Reply: Comma deleted.
Comment 30: -Lines 478-479: “in order to develop different high-yielding and 478 quality new oil palm varieties for planting in” rewrite this part of the sentence
Reply: The sentence were written as in order to develop diverse high-yielding and novel oil palm varieties for planting in….”
Comment 31: -Line 482: “stressful” use “marginal” instead
Reply: Thanks for your suggestion. Correction were made.
Comment 32: -Lines 449-488: Too big paragraph, please split this long paragraph into two medium paragraphs that has smooth paragraph structure
Reply: Suggestion were made.
Comment 33: -Line 490: In Figure 2, Antioxidant enzyme is repeated in the box of biochemical changes, please crosscheck and correct it accordingly
Reply: Changes completed.
Comment 34: -Conclusion is missing, although authors have provided a sentence or two about some interpretations, but largely the conclusion is missing hence please provide a brief paragraph summarizing the key take home messages.
Reply: Conclusion were included.
Comment 35: -References: There are 11 self citations out of 94, which is almost 12% of total references. It is recommended that authors only cite their references where they are relevant and deemed fit.
12, 21, 28, 37, 38, 41, 45, 50, 51, 70, 77,
Reply: Few of the reference were delete.
Round 2
Reviewer 3 Report
Authors have revised as suggested and can be considered further if the Editor accepts the revised manuscript.
Author Response
Dear Reviewer,
Greetings of the day.
Thank you for the favorable comments and suggestions for our manuscript. Careful modifications and improvements were made. The manuscript has been revised for better readability according to your suggestions. Thanks again to the reviewer who helps us to improve the manuscript. We hope that the manuscript is now acceptable for publication.
Thank You

This manuscript is a resubmission of an earlier submission. The following is a list of the peer review reports and author responses from that submission.
Round 1
Reviewer 1 Report
The title misled me to have a different image about the content of the review. "resistance" made me imagine mechanisms how to resist against unfavorable cold, disease or drought attack as results of climate change. But I had to be cautious. Resistance did not always mean such mechanisms in authors' usages. Might be what happened as a result of a sensitive reaction, not a resistant reaction. At least authors should discriminate what they observed was sensitive or resistance reaction of palm plants under such conditions. I recommend the authors to reconsider what they would like to review and insist on in this ms.
Also the title did not include information what kind of stress they are discussing under the term "resistance". "problems and prospects" will describe human-side issue or sensitivity of oil palm against the stresses?
The content looked so scattered and just descriptive in an unorganized order.
L.39 what does "efficient" oil crop mean ? In terms of time or labor?
L.50 conductive?
L.65-70 Maybe such geological issues should be described concisely. An atlas or something similar might simplify the issue.
Also there are many places where English expression should be edited.
References 13 and 18 are identical (Lines 464 and 477). The authors should prepare the ms in a more cautious way.
Reviewer 2 Report
The manuscript reviews the oil palm resistance underlying causes, proposing potential candidates to identify resistant varieties/cultivars/genotypes that assist on maintaining the oil palm world wide needs.
However when reading the manuscript, too details from the literature are given, but not ellaborate summaries and clear conclusions are stated.
One of my main concerns is that is very frequent to read "(...) could be a cold/drought resistance physiological/hormonal/structural indicator", but no further ellaboration on the assumption. As long as it´s a review, I would ask for a further integration of the information to ellaborate how this indicator might work, and which results would lead to select a plant as cold resistant. In general every single gene/protein/hormone that modifies their expression is proposed as indicator, but to me there are 2 different topics: how the cold affects the plant, and how the resistant is acquired, and there´s no clear evidence of tolerance acquisition througout a review that on its title indicates is focused on oil palm resistance.
My second concern is that too much details on papers are given, so the thread/flow on the tolerance topic is getting lost. The numbers of clones, or progenies, or detailed data are not really relevant for stating the output of the paper selected, so this would need an effort on rewritten the masnucript, that I encourage to do, as it´s clear the authors are updated and aware of the state of knowledge ont he topic proposed.
I am also missing a scheme showing the different strategies for approaching the oil palm tolerance, and even more relevant, a table summarizing which paper is focused on what, so the review will be much more comprehensive.
For more details, please checked revised version of the manuscript (Attached).
After considering my revision, I would Reject the manuscript for its publication on its current format. After a deep revision of the document, it might be suitable for publication, but not under the present form.
